# Host- and Microbiota-Derived Extracellular Vesicles, Immune Function, and Disease Development

**DOI:** 10.3390/ijms21010107

**Published:** 2019-12-22

**Authors:** Laurence Macia, Ralph Nanan, Elham Hosseini-Beheshti, Georges E. Grau

**Affiliations:** 1Charles Perkins Centre, The University of Sydney, NSW 2006, Australia; ralph.nanan@sydney.edu.au; 2School of Medical Sciences, Faculty of Medicine and Health, The University of Sydney, NSW 2006, Australia; elham.beheshti@sydney.edu.au; 3The University of Sydney, Sydney Medical School Nepean, Penrith 2751, Australia; 4Vascular Immunology Unit, The University of Sydney, NSW 2006, Australia

**Keywords:** inflammation, immune function, diseases, gut microbiota derived extracellular vesicles, host derived extracellular vesicles

## Abstract

Extracellular vesicles (EVs) are blebs of either plasma membrane or intracellular membranes carrying a cargo of proteins, nucleic acids, and lipids. EVs are produced by eukaryotic cells both under physiological and pathological conditions. Genetic and environmental factors (diet, stress, etc.) affecting EV cargo, regulating EV release, and consequences on immunity will be covered. EVs are found in virtually all body fluids such as plasma, saliva, amniotic fluid, and breast milk, suggesting key roles in immune development and function at different life stages from in utero to aging. These will be reviewed here. Under pathological conditions, plasma EV levels are increased and exacerbate immune activation and inflammatory reaction. Sources of EV, cells targeted, and consequences on immune function and disease development will be discussed. Both pathogenic and commensal bacteria release EV, which are classified as outer membrane vesicles when released by Gram-negative bacteria or as membrane vesicles when released by Gram-positive bacteria. Bacteria derived EVs can affect host immunity with pathogenic bacteria derived EVs having pro-inflammatory effects of host immune cells while probiotic derived EVs mostly shape the immune response towards tolerance.

## 1. Introduction

Communication between cells and between organs is tightly regulated and its disruption can contribute to disease development. Communication can either occur by cell-to-cell contact or by means of soluble factors. Cytokines are by definition soluble factors produced by cells (beyond immune cells) ensuring cell survival, proliferation, differentiation, and/or activation. These factors can act on the cell involved in their production (autocrine), on cells in their close environment (paracrine), or distally. Hormones are produced by endocrine organs as well as immune cells and have mostly distal effects from their sites of release. Both cytokines and hormones bind specific receptors on target cells, which then trigger a signalling activation cascade that will affect the cell phenotype. Both cytokines and hormones have been studied for decades and are the basis of modern physiology.

However, more recently, another process also involved in this type of communication has been highlighted. On top of producing specific factors, cells can send a package of information that is enclosed by cell membrane. This package is also called extracellular vesicles (EVs) and can contain DNA, RNA, proteins, sugars, and lipids. EVs can be produced by any cell type, highlighting their importance throughout evolution from unicellular organisms to metazoans. During the last decade, considerable efforts have been made to investigate the role of EVs in cell communication and disease development. In addition, numerous groups have investigated EVs for their potential as biomarkers and as targeted therapeutic carriers. However, there are major limitations in these areas that need to be addressed in the future. These limitations include a lack of standardized methodology for the isolation and purification of different classes of EVs, inadequate terminology, numerous technical challenges in EV characterization and quantification, and an incomplete identification of the cellular origin of EVs. Historically, most groups have only focused on one or two types of EVs, but have not investigated the respective contribution of all EV subclasses in their model. These are only a few major obstacles that need to be overcome to advance this field. In this review, we discuss the current knowledge on the production and mechanisms of action of EVs in health and disease as well as the impact of commensal and bacteria derived-EVs on cell function and their impact on health and disease.

## 2. Host Cell Derived-Extracellular Vesicles in Health and Disease

### 2.1. Extracellular Vesicles 

EVs are membrane-enclosed vesicles that are released from all cell types. Their importance lies in a unique characteristic, the ability to transfer information to other cells and influence their function [1,2,3,4]. Differences in the array of EVs depend on their cellular origin, biogenesis, and mechanisms of release. EVs are classified into three subcategories based mainly on size, namely, exosomes, microvesicles—previously called microparticles—, and apoptotic bodies. Even though vesicle size is still one of the main factors used in EV categorisation, recent evidence indicates that different classes of EV, and different populations within one EV class, may harbour distinct molecular cargo and play specific functions.

#### 2.1.1. Microvesicles 

Microvesicles (MVs) are plasma membrane-derived, 100–1000 nm diameter vesicles, released by vesiculation from eukaryotic cells. MVs originate as a consequence of the loss of phospholipid asymmetry associated with cytoskeletal remodelling and an increase in cytosolic calcium. Phosphatidylserine and other negatively charged phospholipids, normally sequestered in the inner leaflet of the plasma membrane, redistribute to the outer leaflet. MVs are distinct from other EVs (e.g., exosomes and apoptotic bodies) by virtue of the expression of phospholipids and proteins on their surface [5]. Initially described as “platelet dust” [6], MVs were first studied in the context of coagulation, notably because of the importance of membrane phospholipids [7] and their platelet origin [8,9,10]. The link between MV overproduction and inflammatory processes was suggested by the demonstration of increased MV release upon stimulation by TNF [11]. A role for MV overproduction in disease pathogenesis was further substantiated by the finding of a complete protection against cerebral malaria in ABCA1 knock-out mice [12].

In addition, MVs have been shown to transport RNA between cells [13] as well as molecules present in the plasma membrane, for example, Abcb1 (Pgp-1), which results in the transfer of multi-drug resistance [14].

#### 2.1.2. Exosomes

Exosomes are a nanoscale subtype of EVs with a cholesterol and sphingolipid-rich membrane that is characteristically 30–150 nm in diameter and secreted from all cell types [15]. Although the underlying mechanism of production is not fully understood, it is known that exosomes are formed from the inward budding of the endosomal compartment of cells, resulting in the production of small intraluminal vesicles, followed by the fusion of multivesicular bodies with cell plasma membrane and the secretion of intraluminal vesicles, now called exosomes, into an extracellular matrix. The content of exosomes has been extensively studied for biomarker discoveries. It is known that exosomes contain various classes of proteins, structural and bioactive lipids, different types of nucleic acids (mRNA, microRNA, and DNA), and metabolites. While there are some similarities in the composition of exosomes derived from various cell types, it is now very well established that exosomal cargo is mainly a reminiscent of their parental cell.

While different types of mechanisms have been proposed for exosome uptake, it is known that exosomes interact with the recipient cell, and transmit information resulting in changes in surrounding cells via (1) direct stimulation by membrane ligands; (2) receptor transfer between the donor cells and recipient cells; (3) transfer of genetic information; and (4) direct stimulation by endocytically-expressed surface receptors in recipient target cells [16]. Recently, a thorough description of their protein composition has led to an improved classification of EVs [17].

Accumulating evidence suggests that induction of intracellular calcium [18,19], overexpression of Rab7a, Rab11, and Rab35 or citron kinase [20], reduction in membrane cholesterol, inhibition of cholesterol biosynthesis [21] and Smpd2 (sSMase) [22], and the activation of diacylglycerol kinase α (DGKα) [23] could positively or negatively affect the release of exosomes into the microenvironment.

Once released, exosomes will interact with and are internalized by, recipient target cells via different mechanisms including fusion with the plasma membrane or adhesion to corresponding receptors on the plasma membrane, phagocytosis, micropinocytosis, and lipid raft mediated endocytosis [19,24]. Regardless of their mode of interaction with the recipient cell, exosomes have been shown to have a significant impact on their recipients. From embryonic development to physiological condition, and from tumour growth [25] to tumour vaccine [26], exosomes are proving to be unique as highly mobile, discrete packages of proteins and nucleic acids that are essential for intercellular communications. The timeline of exosome characterisation has been recently reviewed [27].

EVs also include much larger (1–5 μm diameter) vesicles, originating from the shedding of membrane blebs, named apoptotic bodies. Furthermore, oncosomes are EVs produced specifically by malignant cells, and therefore these two groups will not be discussed here [28,29].

### 2.2. EVs in Pathogenesis

After MVs were found in dramatically elevated numbers in cerebral malaria patients [30], the finding that the ABCA-1 knock-out fully protected mice against cerebral malaria pointed towards a pathogenic role for MVs. Remarkably, these mice did not develop any neurological lesions, despite levels of parasitaemia identical to those of wild-type animals [12]. Subsequently, MVs have been shown to transfer antigens [31], activate T cells [32] and monocytes/macrophages [33], alter microvascular endothelial permeability [34], and trigger various signalling changes [32,35]. Importantly, the inhibition of MV production can prevent neurovascular pathology [36,37]. However, direct evidence for a pathogenetic effect of MV was only provided by in vivo transfer experiments in murine cerebral malaria [38].

This array of evidence suggests an interesting parallel with Koch’s postulates [39], supporting the idea that MVs are important players in pathogenesis: indeed, (1) MV abnormal levels were identified at the time of cerebral signs; (2) blocking their production prevented pathology and death; and (3) injecting them reproduced pathology (brain haemorrhages) in recipient animals. Postulate No. 4 (“The microorganism must be reisolated from the inoculated animal”) was not applicable. The involvement of EVs in the pathogenetic mechanisms of disease has been reviewed recently, with particular attention to their effect on microvascular endothelial cells [4].

MVs can show contrasting effects in disease: they can also be potentially protective in sepsis. On one hand, MVs that are found in high plasma levels in sepsis, may be beneficial in early sepsis, given that activated protein C bound to endothelium-derived MV retains anticoagulant activity, and increased circulating MV are protective against vascular hyporeactivity. Elevated MV levels in early sepsis may therefore compensate for the host’s systemic inflammatory response [40]. Additionally, MV levels and MV-cell conjugates predict a favourable outcome [41]. On the other hand, MVs in sepsis carry tissue factor (TF) and contribute to the procoagulant state [8], and platelet MVs increase monocyte binding to endothelium [42].

The characterisation of EVs in clinical settings involves techniques that include flow cytometry, nanoparticle tracking analysis, proteomics, sequencing, lipidomics, and, more recently, various types of vibrational spectroscopy (VS). Compared to traditional omics approaches vs. providing direct information on changes in cells, EVs, and cell medium in one experiment; it avoids complex sampling that can alter molecules unlike proteomics or lipidomic procedures, it detects semi-quantitative changes in lipids, proteins, DNA, RNA, organophosphates, or carbohydrates, and is sensitive to biomolecule conformations (e.g., changes in α-helical and β-sheet proteins [43]).

### 2.3. EVs as Biomarkers

In cerebral malaria patients, the clear association between high MV levels and the occurrence of the neurological syndrome as well as the normalization of these levels at the time of follow up, suggest a potential value of MVs as biomarkers of disease severity [30]. More recently, proteomics analyses of MVs produced in experimental cerebral malaria have identified two elements in MV cargo: carboanhydrase 1 (CA-1) and S100A8, which may be important and linked to pathogenesis as they were specifically increased at the time of the neurological syndrome; these molecules may be considered as biomarkers of severity in malarial infection [44].

A number of diseases including cancer have since been studied along these lines, as reviewed elsewhere [45,46]. Neurological diseases have been widely studied with an attempt to define the biomarker potential of EVs, notably in multiple sclerosis [47,48] and neurodegenerative diseases [49] as well as in stroke and traumatic brain injury [50]. In cancer, EVs are packed with tumour information, as detailed recently [51]. The immunomodulatory properties of EVs have been thoroughly investigated in the context of infections and cancer, as reviewed recently [35,52,53,54,55].

### 2.4. EVs in Homeostasis

All types of EVs, in addition to their increasingly recognised roles in pathogenesis, play important roles in homeostasis. One of the first direct demonstrations of a physiological role for MVs was provided by elegant works from Denisa Wagner’s team, indicating that platelet MVs are crucial in initiating clot formation [56]; the multiple physiological roles of exosomes, notably in immune homeostasis have been reviewed recently [57]. Another area in which EVs are likely to play important homeostatic roles is pregnancy. Pregnancy and lactation represent very unique situations involving the intimate interplay of two distinct eukaryote organisms and two prokaryote microbiomes, namely the mother and her offspring, and both their respective mucosal microorganisms. Effective cross-talk between these different systems has evolved through natural selection to improve reproductive fitness by accommodating the adaption of both the mother and her offspring to their common environments. So far, we have rather limited evidence of these interactions and the best investigated is the role of offspring derived EVs. These have been shown to play a possible role in the development of immunological tolerance between the mother and the foetus. During pregnancy, EVs of foetal origin shed by syncitiotrophoblasts are released into the maternal system. There, these EVs exhibit tolerogenic activities by suppressing NK and T cell responses [58,59] and induce apoptosis through FAS ligand activation [60]. In preeclampsia, a serious pregnancy related condition believed to be in part associated with a breakdown in tolerance between the mother and the foetus [61,62], the release of syncytial derived EVs is increased and appears to contribute to a proinflammatory milieu and vascular complications associated with this condition [58,63]. What makes these EV pro-inflammatory has not been elucidated, but in this context, it has been speculated that their skewing to an anti-inflammatory activity might be a therapeutic option to protect against the development of preeclampsia [64].

EVs crossing the placenta from the maternal to the foetal side have also been implicated in modulating and programming foetal immune responses, but the evidence so far is patchy and indirect. Here, the distinction between microchimerism through maternal cell transfer or by means of EVs remains elusive. It has been shown that maternal alloantigens cross the placenta [65] and potently induce foetal T regulatory cells [66]. Whether this transfer occurs through intact maternal cells or through EVs remains to be shown, but the latter seems to be a reasonable assumption as intact cells might be more susceptible to mechanical and immunological foetal barriers [67]. In this context, it is also of interest to determine whether EVs derived from the maternal microbiome can cross the placenta and exhibit immune modulating properties. We can speculate that maternal microbiome-derived EVs prepare the foetal immune system to facilitate adaptation to its future environment, but evidence to prove or disprove this is lacking.

Finally, during lactation, the secretion of EVs through breastmilk are believed to influence mucosal immunity and increase the gut microbiome diversity of the offspring [68]. EVs from breastmilk are rich in mi-RNA [69,70] and hence may also influence mucosal immunity and the foetal microbiome composition. Whether EVs from the breastfeeding infant or associated microbiome influence lactation or the maternal microbiome is possible, but needs to be further investigated.

While the release of EVs by human eukaryotic cells has been extensively described in health and disease, recent evidence has shown that at least 50% of cells in humans are in reality bacteria, mostly located in the gut. How EVs from commensal bacteria, and from bacteria at large, impact the host will be discussed in the next section.

## 3. Gut Bacteria, Extracellular Vesicles, and Immune Function

### 3.1. Gut Microbiota and Host Physiology

Over the last 20 years, extensive research has demonstrated the key role of the gut microbiota on host physiology and immunity [71]. Analyses of germ-free mice, devoid of gut bacteria, have established the role of gut bacteria on both the development and functionality of immune cells [72]. Germ free mice are characterised by the atrophy of lymphoid organs, lymphopenia, and decreased levels of IgA [73]. They also develop exacerbated diseases such as colitis or allergies [74] or improved disease severity, as shown in the model of gout [75]. More physiological models comparing mice with different gut microbiota have highlighted that beneficial reshaping of the gut microbiota can affect host physiology and could play a role in health and diseases [76]. Indeed, a diet enriched in dietary fibre, which favours the growth of beneficial bacteria, has been shown to protect mice from dextran sodium sulfate (DSS) induced colitis [77], asthma [78], food allergy [79], and type 1 diabetes [80]. In contrast, dysbiosis, detrimental changes in gut microbiota resulting from a consumption of a diet deprived in dietary fibre has detrimental effects on host immunity and favours disease development [81]. The gut microbiota interacts with the host via two main mechanisms: (1) through the release of metabolites and (2) by activating innate receptors. Short chain fatty acids (SCFAs) are the bacterial metabolites that have been the most extensively described [82,83]. While initially they were only considered as bacterial waste products, it is now established that SCFAs play a key role in the differentiation and function of most immune cells including anti-inflammatory regulatory T cells (Treg) [84,85], CD8^+^ T cells [86], macrophages [87,88], tolerogenic CD103^+^ dendritic cells [79,89], and B lymphocytes [90]. SCFAs might affect host cells either through the binding of specific G-protein coupled receptors [88], epigenetic modifications by inhibiting histone deacetylase activity [84], or by directly modulating the metabolic activity of the cells [86,90]. Another mechanism of interaction between the gut bacteria and the host involves the recognition of bacterial membrane components by pattern recognition receptors (PRR). The binding of Toll-like receptor (TLR) 2 by the capsule component polysaccharide A of *Bacteroides fragilis* was shown to promote Treg differentiation [91]. Similarly, we found that stimulation of MyD88, a factor downstream of Toll-like receptor activation, was critical for communication between the gut microbiota and its host. This was demonstrated when the beneficial reshaping of the gut microbiota by dietary fibre failed to protect *Myd88*^−/−^ mice from food allergy [79]. Mice deficient for one of the SCFA receptors GPR43 and GPR109a were also not protected from food allergy under high dietary fibre feeding conditions [79], suggesting that both pathways are important. The complex host–microbiota interaction involves both host sensing of bacterial metabolites, but also a direct interaction with the bacteria. This last point is particularly puzzling, as most bacteria will be physically separated from the host by the mucus layer. Moreover, the effects of live bacteria are often different from those of heat killed bacteria, suggesting that the recognition of bacterial membrane components is more than just a passive interaction. The study by Shen et al. was one of the first to shed light on this phenomenon by showing that commensal bacteria could produce EVs. They showed that administration of EV isolated from *Bacteroides fragilis* could mimic the benefit of administration of the bacteria itself, both promoting Treg development [92]. This finding, soon followed by others, opens a new dimension in our understanding of how gut bacteria affect host homeostasis, and importantly, in the understanding of the systemic and distal impact of gut bacteria on the host. In this part of the review, we will discuss our current knowledge on the impacts of bacteria-derived EVs on host immunity and physiology.

### 3.2. Bacteria-Derived EVs

Like any organisms, bacteria produce EVs as a system of communication with their peers. Contrary to eukaryotic EV, bacterial-derived EV sizes range exclusively below 300 nm in diameter [93]. EVs can promote a selective advantage through the horizontal transfer of genes of resistance to antibiotics to surrounding bacteria. EVs are also an efficient system of detoxification, facilitating the evacuation of toxic compounds important for bacteria survival. Bacterial EVs also favour bacterial adaptation to a novel niche, with a study suggesting that EVs from commensal bacteria could also contribute to their colonization in the gastrointestinal tract [94]. Based on their membrane properties, bacteria are classified as Gram-negative (G−) or Gram-positive (G+). G− bacteria are characterised by a double plasma membrane layer separated by the periplasm. Most EVs produced by G− bacteria secrete outer membrane vesicles (OMVs), which bleb from the outer membrane and contain periplasmic contents such as lipids, outer membrane proteins, lipoproteins, and lipids [95]. Some pathogenic G− bacteria can also naturally produce another type of EV called outer inner membrane vesicles, containing both periplasmic and cytoplasmic components and are thus enriched in DNA or ATP, cytoplasmic, and inner membrane proteins [96]. Whether the outer inner membrane vesicles are produced by commensal G− bacteria has not been reported. There are three models attempting to explain OMV production, as described in detail elsewhere [97], which all suggest that OMV production is a strategy to maintain outer membrane homeostasis. According to these three models, OMVs are produced when the outer membrane lipid asymmetry is compromised (model A), misfolded proteins accumulate in the outer membrane (model B) or when lipopolysaccharide (LPS) is modified (model C) [97]. Recently, explosive cell lysis conditions have been shown to increase the production of OMVs by *P. aeruginosa* [98]. Most studies looking at the impacts of environmental factors on OMV release focused on pathogenic bacteria have identified that elements that would affect the outer membrane integrity such as the nutritional environment, the presence of antimicrobial peptides either host-derived, or antibiotics as well as stressors such as temperature and oxygen availability can promote OMV release [97]. Proteomic analysis of OMV membrane components has shown that membrane proteins can be sub-classified into six categories based on their function: (1) bacteria survival; (2) nutrient sensing; (3) ABC transporters; (4) killing competing bacteria; (5) virulence towards host cells; and (6) modulation of host immune function [99]. OMVs might enter both phagocytic and non-phagocytic cells. Their endocytic entry in the latest, well described in epithelial cells, occurs through their fusion with lipid rafts, micropinocytosis, clathrin or caveolin-mediated endocytosis or following their interaction with host receptors such as TLR2 [95].

The identification of EVs derived from G+ bacteria called “membrane vesicles” (BMVs for bacterial membrane vesicles to discriminate them from host MVs) was achieved in 1990 [100], which is recent when compared to the discovery of OMVs. The membrane of G+ bacteria is dramatically different from that of G− bacteria, which consists of one thick layer of peptidoglycan. How BMVs can make their way through this wall is still not fully understood. Among the current hypotheses, there is the possibility that either increased pressure in the cell would force the passage of BMVs through pores in the wall as well as the creation of pores by cell wall modifying enzymes that could explain the production of BMVs [93]. Like with OMVs, changes in plasma membrane homeostasis can modulate the release of BMVs. Increased membrane fluidity in *S. aureus* when high levels of phenol-soluble modulin are present, increased the release of BMVs [101]. Similarly, antibiotics are known to promote BMV release by cross-linking plasma membrane peptidoglycans [102]. As BMVs have been more recently identified, other factors involved in their production have yet to be discovered. Like OMVs, BMVs carry information from their mother cell such as enzymes, DNA, and RNA that can affect host cells [93].

### 3.3. Pro-Inflammatory Effects of Pathogenic Bacterial EV on Host Immune Cells

Bacterial EVs can interact with host cells through three mechanisms: (1) direct activation of host receptors; (2) delivery of bacterial EV content; and (3) full incorporation of EVs into the host cell cytoplasm [103]. Activation of host PRR by pathogen associated molecular pattern (PAMP) present in the bacterial EV membrane has been particularly well studied in the context of pathogenic bacteria, much less for commensal bacteria. As most immune cells express PRR, bacterial EVs can have profound effects on the immune function. For instance, activation of B cells by OMVs derived from the respiratory pathogenic bacteria *Moraxella catarrhalis* induced B cell activation. Both TLR2 and TLR9 on host B cells, respectively, were activated by the superantigen *Moraxella* immunoglobulin D binding protein (MID) and by the bacterial motif CpG. As a result, B cell activation markers were increased as well as B cell production of IgM and IL-6. In this context, OMVs were used as a decoy to redirect the immune response [104]. OMVs produced by *Porphyromonas gingivalis, Treponema denticola,* and *Tannerella forsythia* have been shown to be phagocyted by monocytes and macrophages and activate NF-κB downstream of TLR signalling. Interestingly, while the three types of OMVs increased pro-inflammatory cytokines (TNF, IL-8, and IL-1β), *P. gingivalis* also triggered the release of anti-inflammatory cytokine IL-10 [105], suggesting that OMVs are not purely pro-inflammatory. Activation of TLR4 on dendritic cells by OMVs increased their expression of activation markers as well as the production of proinflammatory cytokines such as IL-6 and IL-1β [106]. Both studies reported that OMVs can activate the inflammasome pathway in macrophages and dendritic cells. NOD1 is also an important sensor for OMVs in host cells. OMVs can enter non-phagocytic cells such as epithelial cells and their peptidoglycan fraction can interact with NOD1. This interaction promotes autophagy and the release of inflammatory cytokines such as IL-8 [107]. Finally, OMVs can also directly modulate host cell activation via the delivery of small RNA. sRNA52320, which was identified in OMVs isolated from *P. aeruginosa* that could reduce the inflammatory response of airway epithelial cells as well as neutrophil activation and infiltration in mouse lungs [108]. The influence of sRNA in bacteria derived EVs on host cells has been reviewed elsewhere [109].

Based on their immunomodulatory properties, the potential use of bacteria derived OMVs as therapy is more and more investigated, particularly in the context of vaccination. The results are promising as OMVs can boost the generation of CD8^+^ T cells directed towards a specific antigen by antigen cross-presentation by dendritic cells [110].

While less studied, BMVs can also affect immune cell function. EVs derived from *S. aureus* were shown to initiate inflammation by increasing the expression of adhesion molecules (E-selectin, VCAM-1, and ICAM-1) and of IL-6 on skin dermal endothelial cells, increasing the recruitment of monocytes in vitro. Interestingly, this effect was specific for *S. aureus* BMVs and not of G+ BMVs in general, as BMVs derived *Propionibacterium acnes* did not have such an effect [111]. BMVs produced by *Streptococcus pneumoniae* were shown to increase activation markers such as CD86 on human monocyte-derived dendritic cells as well as promote their release of both pro-inflammatory cytokines (IL-8, IL-6, and TNF) and IL-10 [112]. To conclude, EVs produced by pathogenic bacteria can affect host immunity and contribute to pathogenesis.

### 3.4. Tolerogenic Effects of Commensal and Probiotic Derived EVs

Like pathogenic bacteria, EVs derived from gut commensal bacteria can also modulate host immunity. Probiotics are defined as live bacteria with health benefits and some commensal bacteria have probiotic traits promoting host health [83]. While EVs from pathogenic bacteria either trigger inflammation or are used as a strategy to escape the immune response, both OMVs and BMVs derived from probiotics have anti-inflammatory effects and promote immune tolerance. OMVs derived from the probiotic strain *E. coli Nissle* were shown to increase PBMC production of both pro- and anti-inflammatory cytokines, but interestingly these OMVs triggered more IL-10 than OMVs from *E. coli* with no probiotic traits [113], suggesting that like probiotics, their OMVs could contribute to tolerance. One particularity for some EVs derived from probiotics is indeed to induce Treg. OMVs derived from *L. rhamnosus* promote tolerogenic dendritic cells and Treg development in vivo in mice. These tolerogenic effects were linked to the interaction of OMVs on C-type lectin receptors (Dectin-1 and SIGNR1) and TLR (TLR2 and TLR9) on dendritic cells [114]. BMVs derived from *Bifidobacterium bifidum* induce the maturation of monocyte derived dendritic cells and their coculture with naïve T cells favoured their differentiation into Treg [114]. BMVs derived from *B. longum* and from *Enterococcus faecalis* are both internalised by mast cells in a temperature insensitive manner, however, only *B. longum* triggered mast cell apoptosis [115]. This interaction is highly specific as neither B cells, T cells, nor eosinophils interact with EVs derived from these two bacteria. The proapoptotic effect of *B. longum* EVs is mediated by the protein ESBP, which once administered intraperitoneally into mice decreases the proportion of mast cells in the small intestine. BMVs from *S. aureus* stimulate the release of IL-6 from keratinocytes and macrophages in vitro while their pre-treatment with BMVs derived from *Lactobacillus plantarum* counteracted this effect, suggesting potential benefits of probiotic derived EVs during infection. Interestingly, probiotic derived BMVs had this effect only in a preventive context as they had no effect in a context of co-stimulation with *S. aureus* MVs [116]. BMVs from *L. planturum* can also potentially increase host immune defence towards pathogens by increasing the expression of anti-microbial peptides cathepsin B and RegIIIγ in colonic epithelial cell line Caco-2 [117]. BMVs from kefir-derived *Lactobacilli* decrease the release of IL-8 of colonic epithelial cells Caco-2 stimulated with TNF. Interestingly, the BMVs were added after the treatment with TNF, suggesting the therapeutic potential of EVs in inflammatory bowel diseases [118]. *Bacteroides fragilis* has been shown to promote Treg cells through the interaction of PSA with TLR2 [91]. Recently, it was demonstrated that PSA can reach host cells via the release of OMVs, which mimicked the health benefit of the parent bacteria through the exact same mechanisms [92]. A recent study has shown that, like *S. aureus*, commensal bacteria *Ruminococcus gnavus* also encoded protein superantigens on their surface, which are potent IgA inducers. In turn, these bacteria were particularly highly coated with IgA in the gut lumen [119]. Whether these superantigens are carried by BMVs is fully unknown, but would illustrate a nice strategy for the host to exclude pathobionts such as *R. gnavus,* which has been linked to colorectal cancer [120].

### 3.5. Role of Commensal- Versus Probiotic-Derived EVs in Disease and Their Therapeutic Potential

Human studies have shown a link between both infectious, non-communicable diseases and changes in the circulating levels of gut bacteria derived EVs. Patients with an altered intestinal barrier have systemic increase of bacteria derived EVs [121]. In this study, it was shown that patients infected with human immunodeficiency virus (HIV) or inflammatory bowel disease patients with intestinal mucositis linked to therapy both had increased circulating OMVs. While these OMVs are likely to be produced by gut bacteria, the authors did not fully prove it. BMVs were also likely present, which was not further investigated [121]. *Akkermansia muciniphila* is considered as a beneficial bacterium living in the mucus layer. *A. muciniphila* has been shown to be beneficial in human reducing fat mass and glycemia in patients with type 2 diabetes. The benefits involved the interaction of *A. muciniphila* outer membrane protein Amuc_1100 on host cell TLR2 [122]. Interestingly, it was recently shown that patients with type 2 diabetes have decreased OMVs derived from *A. muciniphila* in plasma compared to healthy controls [123], suggesting that there might be a link between type 2 diabetes and disrupted interaction host–gut microbiota via OMVs. The release of EVs from the gut microbiota is a dynamic process with changes in the gut microbiota composition following high fat feeding being paralleled with changes in the type of bacteria-derived EVs. Intriguingly, the resulting EVs did not match the microbiota composition with high fat feeding favouring the release of *Proteobacteria* derived OMVs, suggesting that dysbiosis is also associated with the increased activity of pathobionts [124]. Finally, probiotic derived EVs could also be beneficial in cancer with EVs derived from *Lactobacillus rhamnosus GG* having cytotoxic effects on the hepatic cancer cell line HepG2 in vitro [125].

The therapeutic potential of probiotic-derived EVs has been tested in preclinical models exclusively. Most non-communicable diseases have been linked to increased gut permeability, which is thought to contribute to low-grade systemic inflammation and thus potentially to pathologies such as insulin resistance [126]. Commensal bacteria-derived EVs can directly affect gut permeability with oral administration of OMVs from *A. muciniphila* decreasing gut permeability in a mouse model of diet-induced obesity [123]. In parallel, they identified that mice treated with *A. muciniphila* OMVs had decreased body weight as well as improved glucose tolerance. These metabolic benefits were likely to be linked to the beneficial effect of *A. muciniphila* OMVs on the gut barrier integrity, as 6 h post administration, *A. muciniphila* MVs were only detected in the colon and not in the liver, muscle, fat, pancreas, and spleen [123]. Other bacterial EVs have been shown to affect insulin sensitivity with detrimental effects of *Proteobacteria panacis* OMVs. Mice treated orally with *P. panacis* OMVs developed insulin resistance under normal chow feeding conditions. The authors identified that *P. panacis* OMVs could directly impair insulin signalling in adipocytes and myotubes in vitro. The biodistribution analysis of these OMVs by live imaging revealed that 12 h post-gavage *P. panacis* OMVs were detectable in liver, white adipose tissue, and skeletal muscles, suggesting direct effects of OMVs on insulin sensitive tissues [124]. It is interesting to note that EVs derived from different gut bacteria have different biodistribution in the host, highlighting the complexity of the gut bacteria–host interactions.

Effects of bacteria-derived EVs have also been observed in atopic diseases, with oral administration of MVs from *Lactobacillus planturum* shown to prevent atopic dermatitis development. Of note, administration of 100 μg of *L. planturum* BMVs was as potent as dexamethasone in the prevention of epidermal thickening [116]. Benefits of BMVs in atopy are beyond dermatitis with, as previously mentioned, the protective effects of BMVs derived from *B. longum* in food allergy through their pro-apoptotic effects on mast cells [115]. Finally, the benefits of bacterial EVs on the host have also been reported in inflammatory bowel diseases with oral pre-treatment of mice, with OMVs derived from *B. fragilis* protecting them from trinitrobenzene sulphonic acid (TNBS) induced colitis [92]. Induction of Treg following the interaction of OMV polysaccharide A with TLR2 on tolerogenic dendritic cells was behind these benefits. Similarly, BMVs derived from *Lactobacillus kefir*, *Lactobacillus kefiranofaciens*, and *Lactobacillus kefirgranum* protected mice from TNBS induced colitis by primarily targeting enterocytes [118]. Altogether, these studies highlight the beneficial potential of probiotic-derived EVs in non-communicable diseases. Table 1 summarizes the impact of bacterial EVs on host immunity and health.

## 4. Conclusions

Production of EVs by both eukaryotic and prokaryotic cells is linked to changes in both plasma and intracellular membrane homeostasis, showing that similar mechanisms for the EVs released are highly preserved throughout evolution.

Due to the fact that EVs are difficult to investigate, considerably fewer studies have focused on EVs compared to other soluble factors such as cytokines and hormones. The fact that they are important soluble factors in cell communication in both health and disease, as summarized in Figure 1, provides enormous opportunities to investigate their potential as therapeutic targets or diagnostic biomarkers.

Despite significant advances towards understanding the biology of EVs including their formation, secretion, molecular composition, and influence on the recipient cells over the past decade, the field would enormously benefit from novel protocols to more efficiently isolate the different subtypes of EVs.

EVs have recently garnered attention as communication vesicles, intelligently transporting diverse messenger molecules including RNA, DNA, lipids, and proteins and play a critical role in physiology for intra-host–cell communications as well as inter-kingdom communication. The presence and function of EVs should now be taken into account due to their key role on target cells and impact on physiology.

## Figures and Tables

**Figure 1 ijms-21-00107-f001:**
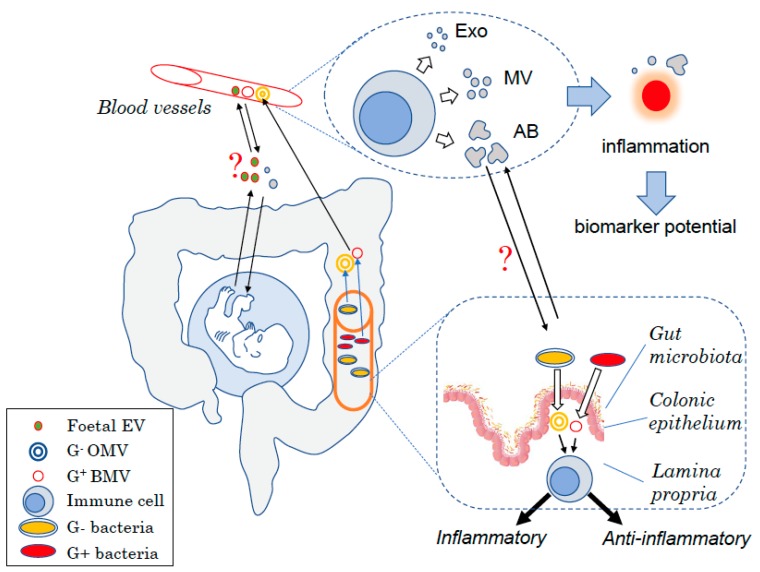
Schematic view of the different types of EVs in humans. EVs are produced at each body site such by mononuclear cells in the blood under the form of exosomes (Exo), microvesicles (MVs), or apoptotic bodies (AB) depending on their size. Changes in the amount of EVs produced have been linked to inflammatory diseases and thus could be used as biomarkers to diagnose diseases. EVs are also key for the host to communicate with endogenous yet “foreign” organisms. During embryonic development, the foetus produces EVs that can interact with cells from the mother and might interfere with maternal tolerance towards the haploidentical foetus. Similarly, the gut bacteria produce EVs that can modulate host–cell immune responses and more broadly the host health status. Bacterial EVs can have pro- or anti-inflammatory effects depending on the bacteria of origin. EVs may also be used for the host to communicate with these “foreign” organisms with maternal EVs that could potentially affect the foetus development as well as host EVs shown to modulate the bacteria composition.

**Table 1 ijms-21-00107-t001:** Bacterial derived extracellular vesicles, immune function, and impact on diseases.

Bacterial Source of EVs	Immune Impact	In vivo Health Outcome	Reference
*Bacteroides fragilis* OMVs	Promote mouse regulatory T cells development via activation of TLR2 on tolerogenic dendritic cells.	Protect from TNBS induced colitis	[89]
*Moraxella catarrhalis* OMVs	Promote IgD and IL-6 release by human tonsillar B cells in vitro and in vivo. This B cell activation involved TLR2 and TLR9 activation as well as increased expression of HLR-DR, CD45, CD64 and CD86.	Contribute to Moraxella sinusitis	[101]
*Porphyromonas gingivalis, Treponema denticola, and Tannerella forsythia* OMVs	Promote human monocytes and macrophages activation as shown by increased production of TNF, IL-8 and IL-1β. *Porphyromonas gingivalis* derived OMV also promoted the release of IL-10. Periodontal OMVs (*Porphyromonas gingivalis, Treponema denticola and Tannerella forsythia*) activate human monocyte and macrophage inflammasome in vitro.	Periodontal diseases?	[102]
*Escherichia coli Nissle* OMVs	Promote production of IL-10, MIP-1α TNF, IL-6, and IL-8 by human PBMC cocultured with Caco-2 cells in vitro.		[113]
*Porphyromonas aeruginosa.* OMVs	Reduce IL-8 secretion by human airway epithelial cells stimulated by LPS in vitro. Decrease the release of KC in bronchoalveolar fluid and neutrophil infiltration in mouse lungs. This was mediated by OMV sRNA affecting target cell gene expression.		[105]
*Lactobacillus rhamnosus* OMVs	Induce regulatory T cells and tolerogenic dendritic cells in Peyer’s patches and mesenteric lymph node in vivo in mice.		[114]
*Staphylococcus aureus* BMVs	Promote the activation of human dermal microvascular endothelial cells in vitro by increasing the expression of E- selectin, VCAM-1 and ICAM-1 and of IL-6 through TLR4 activation and Nf-κB signalling. This activation of microvascular endothelial cells led to increased recruitment of monocytes in vitro.	Might contribute to *Staphylococcus aureus* induced atopic dermatitis	[108]
*Streptococcus pneumoniae* BMVs	Promote the activation of human monocyte-derived dendritic cells by increasing the expression of CD86 and production of IL-8, IL-6, TNF and IL-10 in vitro. BMV also binds human serum complement protein C3, C5b, and factor H, which impair human monocyte phagocytosis of pneumococcal bacteria.		[109]
*Bifidobacterium longum* BMVs	Promote mouse mast cell apoptosis.	Protect mice from food allergy	[115]
*Lactobacillus planturum* BMVs	Decrease IL-6 released by human keratinocytes stimulated by *Staphylococcus aureus BMV in vitro.* Promotes the release of antimicrobial peptide by colonic epithelial cells caco-2 in vitro.	Patients with atopic dermatitis have decreased urine levels of *L. planturum* OMV. Protect mice from atopic dermatitis in vivo.	[116]
kefir-derived *Lactobacilli* BMVs	Decrease the inflammatory response of Caco-2 cells in vitro.	Protect from TNBS inducted colitis.	[118]

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
