# Peer review of "Host- and Microbiota-Derived Extracellular Vesicles, Immune Function, and Disease Development"

_ijms, 2019, doi:10.3390/ijms21010107_

Round 1

Reviewer 1 Report

This article is absolutely interesting and it has a big impact on several fields of medicine and biology.

However, it needs to be much more improved.

This reviewer feels it's too short to be a review, moreover, it does not address all the pros and contras of the fields addressed on the main document.

I suggest rejection with re-submission after an extensive improvement of the text.

A couple of tables would be also useful to compare information and to point out take-home messages for the readers.

I hope to see this manuscript re-submitted to give a certainly more positive evaluation.

Author Response

Thanks for your comments.

To our knowledge there is no limitation on the number of words to be used and we wanted to avoid repeating what others previously reported in reviews.

We do not feel that adding tables would add value to this manuscript.

Reviewer 2 Report

The manuscript by Macia et al., provided a comprehensive review on the complexity of the function and origin of extracellular vesicles in circulation, especially those vesicles released by pathogenic and commensal bacteria.  Two minor things on the manuscript need to be corrected.

The manuscript did not follow the journal manuscript template. The use of gene names in the manuscript did not follow the standard gene symbol, for example pgp-1 in ling 97 should be Abcb1, lines 119-120 Rab7 should be Rab7a, sSMase should be Smpd2.

Author Response

Thanks for your comments.

We have now reformated the manuscript to follow the template.

We have used the standard gene symbol.

Reviewer 3 Report

In this manuscript, the authors summarized the roles of extracellular vesicles secreted by host cells and bacteria. Generally speaking, the review is well organized and covers most of current findings. However, there are still concerns need to be addressed.

Line 132 – 134, authors claimed apoptotic bodies and oncosomes pertain essentially to tumour cell biology. However, apoptotic bodies have been shown important immune regulatory roles (Front Immunol. 2018 Jun 28;9:1486. doi: 10.3389/fimmu.2018.01486). It would be better to rewrite this paragraph. It is hard for the audience to follow “EV”, “Exo”, “MV”, “OMV”, “BMV” or “O-IMV”. I suggest to include a table to explain these terms. Line 221, change “mi-RNA” to “microRNA”. Line 492, change “bacteria” to “bacterial”.

Author Response

Thanks for your comments and we hope that we now addressed the minor concerns.

We have now rewritten the paragraph mentioned as suggested (in red in the text).

We did incorporate a section abbreviations and reduced the number of abbreviation in the text (changes in red).

We did the changes suggested.

Round 2

Reviewer 1 Report

Authors have not improved their work. Paper needs to be improved, as it is not a review in the current state. 

Author Response

We thank the reviewer for his comments. However, we feel that the review in its current form is adequate. This area of research has heavily been reviewed and hence repetitions of already published aspects would be inappropriate. Furthermore, the review with its schematic summary covers the current conceptual understanding of what was initially indicated on invitation to this work. We do not claim that it covers all aspects of EV biology but it certainly covers the scope of what was initially negotiated.

Reviewer 2 Report

The authors addressed my concerns in earlier version. However, tables to summarize different types of EVs released by host cells and microbiome, and the putative functions of different types of EVs would be helpful.

In addition, this is a very controversial field with very little scientific reports; therefore, it is not easy to write a good review on the subject.  The authors might want to add a paragraph to summarize some of the difficulties and challenges in the field.

Author Response

We thank the reviewer for his comments.

As suggested, we have incorporated Table 1, summarizing the different types of bacterial EV and their function.

Concerning a table on the host EV, numerous reviews already have tables defining the different types of host derived EV. Moreover, it is not possible at this stage to attribute a function to a specific type of EV (exosome, MV or oncosome) as the type of cargo and their surface phenotype drive their function.

As suggested, we have added in the text a paragraph highlighting the limitations in the field.

Reviewer 3 Report

Authors have addressed all my concerns.

Author Response

We thank the reviewer.